# Management of Portal Hypertension in Patients with Hepatocellular Carcinoma on Systemic Treatment: Current Evidence and Future Perspectives

**DOI:** 10.3390/cancers16071388

**Published:** 2024-03-31

**Authors:** Valeria De Gaetano, Maria Pallozzi, Lucia Cerrito, Francesco Santopaolo, Leonardo Stella, Antonio Gasbarrini, Francesca Romana Ponziani

**Affiliations:** 1Liver Unit, Centro Malattie dell’Apparato Digerente (CEMAD), Medicina Interna e Gastroenterologia, Fondazione Policlinico Universitario GemelliIstituto di Ricovero e Cura a Carattere Scientifico, IRCCS, 00168 Rome, Italy; valeria.degaetano01@gmail.com (V.D.G.); mariapallozziucsc@gmail.com (M.P.); lucia.cerrito@policlinicogemelli.it (L.C.); santopaolofrancesco@gmail.com (F.S.); leonardo.dr.stella@gmail.com (L.S.); francesca.ponziani@policlinicogemelli.it (F.R.P.); 2Dipartimento di Medicina e Chirurgia Traslazionale, Università Cattolica del Sacro Cuore, 00168 Rome, Italy

**Keywords:** angiogenesis, chronic inflammation, hepatocellular carcinoma, immune checkpoint inhibitors, portal hypertension, tyrosine kinase inhibitors, variceal bleeding

## Abstract

**Simple Summary:**

Portal hypertension (PH) and hepatocellular carcinoma (HCC) are major complications of liver cirrhosis with a detrimental impact on patient outcomes that share common mechanisms. The incidence of PH in patients with advanced HCC is increasing due to the significant improvement of survival outcomes due to systemic therapies, so the management of the complications associated with clinically significant portal hypertension (such as variceal bleeding and ascites) becomes crucial. Moreover, the administration of systemic treatment presents challenges due to the drug-related bleeding risks. This review aims to elucidate the common pathophysiology of PH and HCC, with a specific focus on the influence of systemic therapies on PH. Moreover, we summarize the available evidence regarding PH management in HCC patients. Finally, we discuss potential new strategies for addressing the coexistence of CSPH and HCC.

**Abstract:**

The management of CSPH in patients undergoing systemic treatment for HCC has emerged as a critical concern due to the absence of reliable diagnostic criteria and uncertainties surrounding therapeutic approaches. This review aims to underscore the primary pathophysiological aspects linking HCC and PH, while also addressing the current and emerging clinical strategies for the management of portal hypertension. A review of studies from January 2003 to June 2023 was conducted using the PubMed database and employing MeSH terms, such as “hepatocellular carcinoma”, “immune checkpoint inhibitors”, “systemic therapy”, “portal hypertension”, “variceal bleeding” and “tyrosine kinase inhibitors”. Despite promising results of tyrosine kinase inhibitors in animal models for PH and fibrosis, only Sorafenib has demonstrated similar effects in human studies, whereas Lenvatinib appears to promote PH development. The impact of Atezolizumab/Bevacizumab on PH remains uncertain, with an increasing risk of bleeding related to Bevacizumab in patients with prior variceal hemorrhage. Given the absence of specific guidelines, endoscopic surveillance during treatment is advisable, and primary and secondary prophylaxis of variceal bleeding should adhere to the Baveno VII recommendations. Furthermore, in patients with advanced HCC, refinement of diagnostic criteria for CSPH and guidelines for its surveillance are warranted.

## 1. Introduction

Worldwide, liver cancer is the third most common cause of cancer-related deaths. Hepatocellular carcinoma (HCC) accounts for more than 80% of cases of liver cancer, and its incidence in both men and women has remained stable during the last several years: despite a mild decrease in incidence in East and Southeast Asia and sub-Saharan Africa, a progressive increase has been observed in North America, Europe, and the Middle East [1,2]. Over 80% of cases of HCC occur in patients with liver cirrhosis. Among the underlying liver diseases, viral chronic hepatitis B and C are still the primary causes of liver cirrhosis and HCC (being implicated in 41% and 28.5% of cases, respectively, in 2019), even if the advent of direct anti-viral therapies for HCV and vaccination against HBV will progressively reduce their etiological role. Conversely, the occurrence of HCC related to metabolic dysfunction or alcohol-related liver disease is increasing, particularly in Europe and North America, with an incidence of 5.3% and 6.8%, respectively. The overall survival of cirrhotic patients is strictly influenced by the development of portal hypertension [1,2,3,4].

Portal hypertension (PH) is a major consequence of cirrhosis and, if clinically significant, it is responsible for the development of ascites, bleeding from gastroesophageal varices, and hepatic encephalopathy. Hepatocellular carcinoma (HCC) is another major complication of chronic liver disease. Both PH and HCC contribute to the morbidity and mortality of liver cirrhosis, due to common pathogenetic mechanisms, and each has a dramatic impact on the outcome of the other. In fact, while the presence of PH increases the risk of developing HCC [5,6], abnormal hepatic architecture and vascular invasion may worsen portal hypertension in patients with HCC. Large HCC or multinodular tumors, high alpha fetoprotein (AFP) serum levels, and macrovascular invasion have been associated with more severe ascites [7]. With advances in therapeutic management, the number of patients who present with both PH and HCC is increasing, and physicians face the complexity of managing both conditions.

## 2. HCC and Portal Hypertension: Two Sides of the Same Coin

PH is a clinical syndrome that complicates liver cirrhosis in almost 90% of cases during the natural history of liver disease, and it is defined by an increase in portal venous pressure beyond 10 mmHg [8,9]. Portal venous pressure (P) is proportional to blood flow (Q) and vessel resistance (R), according to Ohm’s law (∆P = Q × R). In normal conditions, a gradient from 1 to 5 mmHg exists between the systemic and the portal venous system, and it is regulated by several substances in response to hemodynamic changes to maintain blood flow homeostasis [9]. Chronic inflammatory conditions such as liver cirrhosis have an impact on portal pressure, acting directly or indirectly on vascular resistance and blood flow [10,11]. In the early subclinical stage, portal venous pressure rises above the threshold of 5 mmHg; as liver cirrhosis worsens, the PH increases above 10 mmHg—a condition called clinically significant portal hypertension (CSPH), which is characterized by the development of signs such as esophageal and gastric varices, ascites, hepatic encephalopathy, and hepatorenal syndrome. Above 12 mmHg, the risk of variceal hemorrhage is high. PH is associated with increased morbidity and mortality, due to hepatic decompensation and the development of HCC [6,12]. Indeed, HCC may arise in a previous condition of PH, accelerating its progression, or it can be involved in the development of de novo PH. 

The most recent guidelines on the management of PH in patients with liver cirrhosis were released during the Baveno VII conference, but these recommendations cannot be fully applied to patients with PH and HCC [9]. In addition, many drugs used in the treatment of advanced HCC may increase the risk of bleeding [13]. This knowledge gap has dramatic consequences for the management of advanced HCC in the presence of CSPH; thus, real-life studies are trying to overcome these concerns. In this review, we describe the mechanisms that lead to PH in patients with HCC, reporting the current data and recommendations on the management of PH during systemic therapy. 

### PH in Liver Cirrhosis and HCC: Physiopathology

HCC and liver cirrhosis share similar mechanisms related to morphological changes in the liver parenchymal architecture that lead to an increase in portal pressure [13].

The first driver of PH in liver cirrhosis is linked to chronic inflammation, extracellular matrix deposition, and the development of regenerative nodules that compress the vascular structures and, in a second step, lead to increased blood flow in the portal venous compartment [14]. The functional vascular unit of the liver, the hepatic sinusoid, is composed of hepatocytes, hepatic stellate cells (HSCs), and liver sinusoidal endothelial cells (LSECs), which are all involved in the complex process of portal pressure homeostasis [15]. Inflammatory cytokines (transforming growth factor (TGF), platelet-derived growth factor (PDGF), tumor necrosis factor (TNF), and interleukin 1 beta (IL-1β) stimulate the activation of HSCs located in the perisinusoidal space of Disse near the endothelium into myofibroblasts, which express alpha smooth muscle actin (SMA), can contract in response to the increase in blood flow [14,15,16] and can also release extracellular matrix (ECM) components, such as collagen type I and II. Kupffer cells and LSECs, in a paracrine way, stimulate activated HSCs to contract [17,18]. LSECs constitute the thin and fenestrated layer of the endothelium that provides nutrients and oxygen to hepatocytes [19]. In healthy conditions, they are not sustained by a basal membrane, so their connection is leaky compared to the capillaries of other body sites. The exposure to inflammatory cytokines transforms the LSEC phenotype, leading to the formation of thicker capillaries, with the loss of the typical sinusoidal fenestration and the production of a basal membrane. Moreover, these vessels become more resistant to vasodilators; together with an exacerbated release of vasoconstrictors, this results in an increase in portal pressure [20,21,22]. As a consequence, LSECs are subjected to shear stress and undergo mechanical damage, with the exposure of the ECM and the activation of subendothelial prothrombotic components. In the presence of chronic inflammation, LSECs enhance the production of von Willebrand Factor (vWF) molecules that combine in high-molecular-weight multimers [23]. Normally, wWF multimers are not present in hepatic sinusoids and do not aggregate in high-molecular-weight multimers due to the action of their natural inhibitor, a disintegrin and metalloproteinase with a thrombospondin type 1 motif member 13 (ADAMTS-13), which is mainly produced by HSCs in healthy conditions but not in the case of its transformation into myofibroblasts [24,25]. Therefore, in conditions of chronic inflammation, the changes of LSEC and HSC phenotype causes an imbalance between vWF and ADAMTS-13 production. As a result, increased vWF deposition in perisinusoidal spaces favors platelet aggregation and fibrin deposition with the formation of parenchymal microthrombi, which stimulates fibrosis deposition and increases vascular resistance [26]. Overall, these alterations worsen local hypoxia, activating hepatocyte damage-associated molecular patterns (DAMPs) and triggering HSCs [17]. HSCs try to revert this process by releasing Vascular Endothelial Growth Factor (VEGF) and PDGF, with the consequence of creating new vessels, which are not sufficient to increase hepatocyte perfusion, thus worsening local ischemia. Similarly, HCC can influence the normal physiology of sinusoidal cells. Some studies, utilizing transcriptomic techniques, identified HSCs as the progenitors of cancer-associated fibroblasts (CAFs) in HCC [27,28,29,30,31].

Interestingly, activated HSCs also have a role in angiogenesis and vessel formation either directly, since they secrete substances such as VEGF and angiopoietin, or indirectly, by the expression of angiogenic receptors that are targets of systemic or paracrine mediators, such as VEGF, PDGF, Interleukin 6 (IL6), and TGF 1 beta [18,32]. Furthermore, in HCC LSECs develop a basal membrane and loose their fenestrae, a process which is similarly observed during PH development in LSECs of patients with liver cirrhosis. With HCC progression, LSECs reduce the expression of intercellular adhesion molecule-1 (ICAM1) and acquire a proangiogenic and prothrombotic profile [19]. Peritumoral endothelial cells also increase the expression of proangiogenic genes, IL6 and its receptor, in response to tissue hypoxia, fostering tumor-induced neovascularization [20], triggering HSC fibrotic activity and release of Hepatocyte Growth Factor (HGF), which leads to nodule formation in hepatocytes [33]. Angiogenesis is another common mechanism shared by liver cirrhosis and tumorigenesis. It leads to the formation of new vessels, driven by tumor cells, to guarantee oxygen and nutrient storage for HCC. In physiological conditions, angiogenesis is regulated by several factors, such as VEGF, FGF, Epidermal Growth Factor (EGF), PDGF, angiopoietins, and several cytokines and chemokines (IL-6, IL-8, TNF alpha, and interferon (IFN)) [34,35]. The pseudocapillarization of sinusoids induced by chronic inflammation reduces the amount of nutrients available to hepatocytes. Inflammation and hypoxia lead to the so-called “angiogenic switch”, which enhances vessel proliferation; the hypoxic environment induces transcription of hypoxia inducible factor 1 a (HIF 1a), which accumulates in the nucleus and dimerizes with HIF 1b to form the HIF 1 complex, causing the transactivation of target genes such as VEGF [34,35]. VEGF binds its receptors located on the endothelial cells and activates a pathway that favors endothelial proliferation and the formation of new vessels [36]. The binding of VEGF/VEGF Receptor (VEGFR) isoform 2 leads to a phosphorylation cascade that activates the phosphatidylinositol 3-kinase/protein kinase B (PI3K/AKT) and RAF/mitogen-activated protein kinase (MAPK) pathways, resulting in endothelial cell activation, proliferation, and organization into neovessels. Tumor vessels are very different from the original ones, structurally and functionally abnormal, and unevenly distributed throughout the tumor, leading to avascular areas. Therefore, hypoxia and lactic acidosis persist during tumor expansion, in a vicious circle, contributing to the maintenance of VEGF production, which is further supported by cytokines and shear stress [34,35,36]. Recent studies have demonstrated that angiogenesis may impact liver fibrogenesis, stimulating HSCs to release ECM components via VEGF. It is to be noted that HSCs have a regulatory role regarding VEGF functions which is not only limited to its secretion but is also related to an increase in the expression of VEGF and angiopoietin receptors and the organization of LSECs into new vessels in a paracrine and autocrine manner in an attempt to lower PH. In addition, HSCs produce Chemokine (C-X-C motif) ligands (CXCLs) 8, 9, 10, and 12, which promote angiogenesis during liver fibrosis. VEGF also enhances the activation of nitric oxide synthase in splanchnic circulation, a condition that favors the development of peripheral systemic vasodilation, increased cardiac output, portal systemic collaterals, and the onset of hyperdynamic circulation [37,38,39]. In this setting, the opening of portosystemic collaterals or pre-existing vessels is a consequence of excessive angiogenesis. Studies have reported that double inhibition of VEGF and PGDF reduces this process and, consequently, the systemic vasodilation associated with liver cirrhosis [40,41,42]. VEGF has a pivotal role in promoting the early stage of tumor neovascularization, while PDGF beta promotes the maturation of new vessels, inducing overall endothelial cell proliferation and migration in peritumoral tissue [43,44]. VEGF levels are also associated with tumor size, microvessel density, tumor invasion of the portal vein, and inflammation [45]. The tumor microenvironment plays a significant role in this process, since HSCs, CAFs, and tumor-associated macrophages are involved. Interestingly, CAFs seem to have originated mainly from activated HSCs in HCC, and they produce cytokines such as IL6, TGF beta, and HGF, promoting angiogenesis [32].

Similarly to cirrhosis, vWF and ADAMTS-13 levels and the impaired physiological functions associated with them seem to be associated with HCC and portal hypertension not only through the stimulation of fibrogenesis, as previously discussed, but also through the induction of angiogenesis, promoting VEGF Receptor 2 phosphorylation, which enhances VEGF expression. Studies have shown that vWF is an independent predictor of HCC occurrence and is associated with tumor mass [46,47], tumor growth, worsening of PHT [48], and metastatic dissemination acting on the STAT-3 pathway [23,49,50]. Another study supports the influence of ADAMTS-13/vWF on hypercoagulability, a risk factor for PH in HCC.

Finally, bacterial translocation from the intestinal lumen is another mechanism that favors the development of PH. Gut bacteria-derived products may stimulate the fibrotic phenotype of HSCs. Moreover, HSCs produce factors such as VEGF and angiopoietin 1, which facilitate the capillarization process of LSECs. Products such as lipopolysaccharides (LPSs) and others derived from gut bacteria are also involved in the onset of systemic hyperdynamic circulation and the dysfunction of extraintestinal organs [51,52]. Figure 1 summarizes the main mechanisms of portal hypertension.

## 3. How Does Systemic Therapy Affect Portal Hypertension?

### 3.1. Systemic Therapy in HCC: State of the Art

The first agents approved for the treatment of advanced HCC were the tyrosine kinase inhibitors Sorafenib, Lenvatinib, Regorafenib, and Cabozantinib, which are oral agents that act primarily on VEGF receptors to mimic VEGF ligands, leading to the downregulation of several proliferative pathways; in particular, they counteract neoangiogenesis. Another similar agent that has been discovered is Ramucirumab, which is a fully human anti-VEGFR-2 monoclonal antibody. Recently, immune checkpoint inhibitor (ICI)-based therapies have demonstrated promising beneficial effects; the combination of Atezolizumab, an anti-PD-L1 agent, with Bevacizumab, an anti-angiogenic agent, showed superiority over Sorafenib [53], and represents nowadays the first-line treatment of choice, together with Tremelimumab/Durvalumab, another regimen which combines two immune checkpoint inhibitors [54]. Other ICIs, such as Nivolumab [55], Nivolumab plus Ipilimumab [56], and Pembrolizumab [57], have been approved by the FDA for the treatment of HCC.

The association of an anti-VEGFR (Bevacizumab) and an ICI (Atezolizumab) is based on the evidence that anti-angiogenic drugs, through the inhibition of abnormal vessel proliferation, can to switch the anergic and immune-tolerant phenotype of the TME into an immune-active one. Tissue hypoxia and acidosis in the TME block T cell effector infiltration through multiple mechanisms, including programmed death ligand 1 (PD-L1) upregulation, increase in T regulatory cells, infiltration of M2-like macrophages, and production of immunosuppressive factors, such as VEGF and TGFβ. These mechanisms are downregulated in the presence of anti-VEGFR agents, even if excessive doses or prolonged treatment duration may reduce tissue perfusion, leading to the upregulation of the hypoxia–neoangiogenetic pathways [58].

Moreover, VEGF is involved in the survival of endothelial cells and the maintenance of the integrity of microvascular architecture [59]. VEGF A effects are mediated by two tyrosine kinase receptors: VEGFR-1 and, mostly, VEGFR-2 [60]. Stimulation of VEGFR-2 results in endothelial cell survival via the expression of the anti-apoptotic proteins Bcl-2a and A1 and the activation of MAP kinases ERK1 and -2, but also their migration, and the production of nitric oxide (NO) and prostacyclin [60].

Hence, a possible consequence of antagonizing the VEGF pathway may be a decreased capacity for the renewal of endothelial cells in response to trauma [61]; this might result in endothelial dysfunction and defects in the inner vascular lining, exposing subendothelial collagen [59,62], as well as decreased matrix deposition in the supporting layers of vessels. Therefore, the flip side of the coin when using anti-VEGF agents is not only a tendency to bleed [63] but also an augmented risk of thrombosis due to tissue factor activation secondary to exposure to subendothelial collagen [62]. For these reasons, anti-angiogenic agents, such as TKIs and Bevacizumab, are contraindicated or should be prescribed with caution in patients with thrombotic or cardiovascular disorders and a history of bleeding. It is also mandatory to assess the risk of variceal bleeding in patients with CSPH when starting therapy with Atezolizumab/Bevacizumab, and it is highly recommended before starting TKIs.

Recent therapeutic advances and access to multiple lines of therapy have led to an improvement in the overall survival of patients with HCC from a median of 10.7 months with Sorafenib to up to 19.2 months with Atezolizumab/Bevacizumab and 16.4 months with Tremelimumab/Durvalumab, understanding the impact of different therapies on portal hypertension and the prevention of cirrhosis complications are new issues to deal with in patients undergoing systemic therapy. As shown in Table 1, the effects of systemic therapy on portal hypertension can be variable among different agents, and more efforts are necessary to better understand the impact of new therapies on portal hypertension, risk of bleeding, and liver decompensation.

### 3.2. Effect of Sorafenib on Portal Hypertension

Sorafenib is a multikinase inhibitor with a dual mechanism of action against tumor cell proliferation and tumor angiogenesis through its action on the Raf/MEK/ERK pathway and the VEGF and PDGF receptor families [64,65]. VEGFR inhibition is widely considered the critical mechanism of action of Sorafenib in several cancer types, such as HCC, renal cancer, and thyroid cancer [66]. As described above, VEGFR targeting agents may increase the risk of bleeding, and this can be a major concern regarding the use of Sorafenib in patients with underlying liver cirrhosis. On the other hand, some preclinical studies have suggested a possible positive effect of Sorafenib on PH in cirrhotic animals. Mejias et al. showed that, following Sorafenib administration, rats with pre- and intrahepatic PH showed an 80% decrease in splanchnic neovascularization and a marked attenuation of hyperdynamic splanchnic and systemic circulations, as well as a significant (18%) decrease in the extent of portosystemic collaterals [67]. Fernandez et al. suggested that the dual inhibition of VEGF and PDGF signaling pathways significantly reduces splanchnic neovascularization and pericyte coverage of neovessels, resulting in a 40% decrease in portal pressure in a rat model [68]. Another study [69] conducted in partial portal vein-ligated rats showed a beneficial effect of Sorafenib on PH, particularly about the formation of portosystemic collateral vessels and portal venous inflow. Although several preclinical studies suggested a possible positive effect of TKIs on PH, whether it occurs in humans remains unclear. A small study on seven Child Pugh A or B patients with advanced-stage HCC and PH treated with Sorafenib (400 mg twice daily) evaluated the changes in portocollateral circulation before sorafenib therapy and at day 30 through cine phase-contrast magnetic resonance imaging velocity mapping. The study showed a decrease in portal venous flow of at least 36% in the Sorafenib group, with reversion to baseline values after Sorafenib withdrawal. In contrast, no specific change was observed in the azygos vein or the abdominal aorta [70]. A pilot study on thirteen patients with advanced HCC and preserved liver function (Child Pugh A or B) investigated the short-term effects of Sorafenib on hepatic venous pressure gradient (HVPG) and systemic hemodynamics, as well as intrahepatic mRNA expression of genes involved in liver fibrogenesis, angiogenesis, and inflammation. The results showed a potential beneficial effect of Sorafenib on HVPG, especially for alcohol-related cirrhosis [71].

### 3.3. Effect of Lenvatinib on Portal Hypertension

Lenvatinib is a multikinase inhibitor targeting VEGFRs 1–3, fibroblast growth factor receptors (FGFRs) 1–4, PDGF receptor-α, RET, and KIT [72]. Its potent inhibition of the FGFR pathway is considered the primary mechanism of action in HCC [73]. Its action on FGF19 and FGF 21, which are known to promote liver regeneration and maintenance of hepatic metabolism, could impact underlying liver disease and favor PH development; however, further studies are needed to address this point [13].

The hemodynamic effect of Lenvatinib appears different from that of Sorafenib; while Sorafenib seems to have a beneficial effect on portocollateral circulation, some evidence suggests that Lenvatinib could worsen PH. In particular, a study including 454 patients treated with Sorafenib or Lenvatinib showed a significant increase in the incidence of hepatic encephalopathy in the Lenvatinib group, especially in patients with poorer liver function (Child Pugh > B7) receiving an off-label treatment [74]. Another study showed a significant reduction in portal venous flow velocity (measured by duplex Doppler ultrasonography) in 28 patients after two weeks of treatment with Lenvatinib, reflecting increased portal congestion [75]. It is interesting to point out that PH may affect the efficacy of systemic therapy. A prospective multicenter study showed that Lenvatinib prolonged progression-free survival in patients with more severe PH and poorer liver function expressed as albumin–bilirubin score (ALBI) grade 2 or 3, reflecting an increased effect related to changes in drug pharmacokinetics and exposure [76].

### 3.4. Effect of Atezolizumab/Bevacizumab on Portal Hypertension

The new standard of care in the first-line therapy for advanced HCC includes an inhibitor of PD-L1 (Atezolizumab), and an anti-VEGF (Bevacizumab). This combination regimen seems not to be associated with significant changes in HVPG, although the measurement can be affected by neoplastic portal invasion or by the presence of large and infiltrative tumors, leading to false negatives [77,78]. This seems to be supported by real-life data, showing good tolerability in both Child Pugh A and Child Pugh B patients [74]. The major concern when using anti-VEGF agents is bleeding; the IMbrave150 study excluded patients with untreated or incompletely treated varices and bleeding or a high risk of bleeding. Compared with the Sorafenib arm, the Atezolizumab/Bevacizumab arm presented a higher risk of bleeding (14% vs. 6%) [79] and, specifically, a higher risk of variceal bleeding (7% vs. 4.5%) [53]. However, the type of bleeding most associated with Atezolizumab/Bevacizumab was epistaxis. A prospective study showed an increased risk of acute variceal bleeding in patients treated with Atezolizumab/Bevacizumab compared to Sorafenib; however, the major risk factor that emerged was a previous history of acute variceal bleeding, while size of varices, ongoing anti-coagulation, and tumor vascular invasion did not appear to be associated with the condition [77]. Thus, in patients with a history of bleeding, a treatment based only on ICIs (e.g.,Tremelimumab/Durvalumab) may be preferred over Atezolizumab/Bevacizumab, which is not contraindicated but requires careful monitoring [77]. Interestingly, a real-life study on the administration of Atezolizumab/Bevacizumab in Child Pugh A or B cirrhotic patients with advanced HCC reported similar rates of bleeding during immunotherapy. Bleeding was not associated with the stage of tumor disease or baseline presence of portal vein thrombosis. However, the severity of bleeding reported was minor, with only nine patients reported to have experienced grade 3 variceal bleeding [79].

Besides the risk of bleeding, acute liver injury after exposure to Atezolizumab has been described [80]. Another case report described a rapid progression of esophageal varices after Atezolizumab/Bevacizumab treatment for HCC, highlighting the necessity of paying attention to detect the worsening of esophageal varices during Atezolizumab/Bevacizumab therapy and poor wound healing after endoscopic variceal band ligation treatment [81].

### 3.5. Effect of ICIs on Portal Hypertension

To our knowledge, no data are available on the effect of ICIs on portal hypertension. A multicenter international observational study investigated whether the use of non-selective beta-blockers (NSBBs), which can modulate PH, could have an impact on patients receiving ICIs. They found no significant association between beta-blocker exposure and overall survival. No significant associations were observed between beta-blocker exposure and secondary outcomes, including progression-free survival (PFS), objective response rate (ORR), and development of adverse events (AEs) [82].

As immune checkpoints have a role in maintaining immune homeostasis and self-tolerance, the modulation of these pathways can result in various immune-related adverse events. A diagnostic challenge when facing multiple signs of organ dysfunction is to discriminate whether they are related to underlying liver disease or immune-related adverse events [83]. Compared with non-HCC malignancies, liver injury related to ICIs is more common in patients with HCC. However, the outcomes in terms of the requirement of corticosteroid therapy, hepatic decompensation, treatment discontinuation, and overall survival are similar [84]. A retrospective case series of patients with advanced HCC and Child Pugh B cirrhosis treated with Nivolumab showed high rates of adverse events mainly associated with liver dysfunction and advanced tumor burden. The overall rates of AEs attributed to Nivolumab were similar to those reported in the Child Pugh A cohorts in the CheckMate 040 trial [85]; the findings were similar in another study [86].

As regards the combination of Tremelimumab/Durvalumab, in the HIMALAYA trial, no meaningful liver toxicity and no treatment-related gastrointestinal or esophageal variceal bleeding was observed; however, the study was predominantly conducted on patients with compensated cirrhosis (Child–Pugh A) [87].

**Table 1 cancers-16-01388-t001:** Effect of systemic therapy on portal hypertension, risk of bleeding, and liver decompensation.

Systemic Treatment	Molecular Target	PH	Risk of Bleeding	Risk of Liver Decompensation/Other
Tyrosine kinase inhibitors				
Sorafenib	VEGFR, PDGFR, Raf/MEK/ERK pathway	↓	↑	↑
Lenvatinib	VEGFR 1-3, FGFR 1-4, PDGFR-α, RET, KIT, FGF19, FGF21	↑	↑	↑ (Encephalopathy)
Regorafenib	VEGFR-1-3, PDGFR, TIE2, FGFR c-KIT, RET, RAF-1, BRAF, V600E BRAF [88]	↓ (Murine model) [89]	No data	Apparently none
Cabozantinib	MET, VEGFR 1, 2, 3, AXL (GAS6 receptor), RET, ROS1, TRKA, TRKB, TYRO3, MER, KIT, FLT-3 [90]	No data	No data	No data
Monoclonal antibodies				
Ramucirumab	VEGFR-2	No data	No data	No data
Immunotherapy				
Atezolizumab/Bevacizumab	PD-L1, VEGF	No data	↑	
Tremelimumab/Durvalumab	CTLA-4, PD-L1	No data	No	IRLI
Nivolumab	PD-1	No data	No data	
Pembrolizumab	PD-1	No data	No data	
Ipilimumab	CTLA-4	No data	No data	

Abbreviations: CTLA-4, Cytotoxic T Lymphocyte Antigen 4; FGFR, Fibroblast Growth Factor Receptor; IRLI, immune-related liver injury; PD-1, Programmed cell death-1; PD-L1, Programmed cell death ligand-1; PDGFR, Platelet-derived Growth Factor Receptor; PH, portal hypertension; VEGF, Vascular Endothelial Growth Factor; VEGFR, Vascular Endothelial Growth Factor Receptor.

## 4. Real-Life Studies and Management of Portal Hypertension in Advanced Hepatocellular Carcinoma: The Baveno VII Criteria and Beyond

As shown in Figure 2, clinicians who deal with patients with HCC have to take several issues into consideration.

The Baveno VII criteria may not be fully applicable in patients with advanced HCC, as the management of PH in this specific setting was not discussed at the conference. In fact, platelet levels may be increased and liver stiffness measurements may be overestimated in the presence of HCC [13,91], making it difficult to rule in/out CSPH with non-invasive criteria, avoiding screening endoscopy. Moreover, HVPG values may be increased in the presence of HCC due to arteriovenous shunting, modification of liver architecture, and possible vascular invasion of the portal vein and/or its branches [77,78].

However, a beneficial impact of NSBBs on survival has been described in patients with HCC, and a significant reduction in the risk of developing this type of tumor has been reported in patients with liver cirrhosis [92,93].

A major concern in patients with HCC is bleeding. A retrospective study showed that HCC patients who experienced variceal bleeding had significantly worse survival than those who did not bleed, reflecting more severe underlying liver disease [94]. Moreover, patients with HCC show higher rates of 5-day treatment failure for acute variceal bleeding, 6-week mortality, and cirrhosis-related complications [7].

Hence, endoscopic assessment of PH is still mandatory in patients with HCC to prevent major complications and mortality related to acute variceal bleeding, with even higher attention being paid during systemic therapy. Starting treatment with NSBBs in patients with HCC and liver stiffness measurements (LSMs) ≥ 25 kPa independently of their variceal status, avoiding upper endoscopy before and during treatment, has also been suggested [78]. However, besides the risk of overtreatment in this group, in all the other cases, upper endoscopy is strongly recommended within six months before starting treatment and then should be repeated annually. Patients should also be monitored carefully for bleeding events during treatment, as they can be life-threatening, regardless of the type of therapy (Atezolizumab/Bevacizumab or TKIs) [95]. An interesting survey [96] on French practice regarding the management of PH in patients with advanced HCC showed significant disparities between oncologists, hepatologists, and gastroenterologists. The survey investigated the different strategies adopted for the screening of PH in cirrhotic patients with and without advanced HCC, the management of acute variceal bleeding, and primary and secondary prophylaxis according to the systemic treatment performed. Significant differences were observed in terms of screening and primary prophylaxis for acute variceal bleeding, with a preference for upper endoscopy, according to the Baveno VI criteria (study conducted from June 2021 to June 2022). The study also highlighted that physicians used endoscopic band ligation as the primary prophylaxis in more than one-third of patients with advanced HCC and large varices, even though it may delay the initiation of treatment and expose patients to ulcer bleeding [97]. Regarding secondary prophylaxis, a combination of NSBBs and banding was the treatment of choice. It may be challenging to decide which strategy to adopt in the case of acute variceal bleeding or when endoscopic band ligation is needed during systemic treatment. Band ligation precludes the use of anti-angiogenic agents and TKIs at least for the following 14 days, and monotherapy with Atezolizumab may be considered; nevertheless, whether to resume the treatment with Bevacizumab depends on multiple factors (the result of variceal therapy and clinical stability) and should be performed after a multidisciplinary discussion [95]. A transjugular intrahepatic portosystemic shunt (TIPS) is another option, but whether it can be applied to patients with HCC has to be defined. TIPS placement has been associated with a poor outcome and a potential increase in the risk of liver failure after locoregional therapies [98]. However, pre-emptive TIPS in the case of acute variceal bleeding may increase overall survival and can allow access to HCC treatments that would otherwise be precluded (especially in the presence of refractory ascites) [99]. An alternative strategy may be the use of ICI-only-based regimens for the treatment of HCC after bleeding or in patients at high risk of bleeding. As shown in the HIMALAYA trial, no bleeding events have been associated with the Tremelimumab/Durvalumab combination [87], nor have bleeding events been reported for Nivolumab [100] and Pembrolizumab plus Lenvatinib regimens [101].

Portal invasion might also precipitate PH and related events. In the case of non-neoplastic portal thrombosis, anti-coagulation is needed to limit its extension and further increase PH. On the other hand, at present, there is no evidence of the role of anti-coagulation in the case of neoplastic thrombosis.

As far as other solid tumors are concerned, anticoagulant therapy, including new oral direct anticoagulants, is not contraindicated in patients receiving Bevacizumab. Hence, in patients with a recent banding, low-molecular-weight heparin may be preferred to facilitate the management of bleeding if necessary [78].

Beyond bleeding events, the treatment of HCC in patients with decompensated cirrhosis (Child Pugh B or higher), remains challenging and is supported by poor evidence, as clinical trials exclude these patients, mainly for safety reasons but also because liver decompensation represents a competing risk for survival [102,103,104]. At present, Sorafenib is the only TKI approved for Child Pugh B patients. The prospective observational registry study Global Investigation of therapeutic DEcisions in hepatocellular carcinoma and Of its treatment with Sorafenib (GIDEON) did not find a significant association between Child Pugh B score and increased incidence of drug-related AEs or of AEs leading to treatment discontinuation [105]. A meta-analysis of 30 studies showed that Child Pugh B liver function is associated with worse overall survival in patients treated with Sorafenib, despite a similar response rate, safety, and tolerability [106]. Similar to Sorafenib, Lenvatinib has shown good efficacy in Child Pugh B patients; however, OS was limited by higher rates of treatment dose reduction or discontinuation, AEs, and liver-related deaths [107]. As regards ICI-based regimens, Atezolizumab/Bevacizumab showed similar rates of treatment-related AEs and bleeding events between Child Pugh A and Child Pugh B patients, with a better OS in patients with compensated cirrhosis [79]. Moreover, an ALBI grade ≥ 2 and a poor performance status represent risk factors for the development of ascites and hepatic encephalopathy. Interestingly, Atezolizumab monotherapy does not appear to be associated with these adverse events [107]. At present, considering the scarce evidence on the safety and efficacy of HCC treatments in patients with decompensated cirrhosis, immunotherapy seems to be the safest option since no significant liver-related AEs have been reported in current studies [53,86,100], but real-world data are needed to confirm this.

## 5. Conclusions

Advances in treatments and prolonged survival of patients with cirrhosis have made the scenario of non-resectable HCC more and more complex, and the simultaneous management of both PH and HCC needs to be dealt with. Complications of PH, such as bleeding varices, ascites, and hepatic encephalopathy, may lead to treatment interruption, with negative consequences for patients’ outcomes and risks of tumor progression. Thus, preventing these events is mandatory. The Baveno VII consensus, unfortunately, has not expressed an opinion on the management of PH in patients with advanced HCC. While waiting for specific indications, treatment should follow the standard recommendations for liver cirrhosis, but awareness of pitfalls, such as overestimation of CSPH and critical management of acute variceal bleeding, in this setting is needed. However, before starting treatment, performing upper endoscopy should be mandatory, and it should be repeated regularly during treatment. Nevertheless, given the high risk of re-bleeding when using anti-VEGFR, at present, there is a lack of evidence about the therapeutic strategy to adopt after acute variceal bleeding or endoscopic band ligation; thus, the role of ICI-only-based therapies should be investigated. Further investigation is needed to define the role of pre-emptive TIPS in well-selected patients in the case of variceal bleeding. The impact of ICIs on PH remains an uncharted territory, but the liver-independent metabolism promises reassuring safety results in cirrhotic patients with mild or moderate decompensation and may offer a new treatment landscape for these patients orphan of therapy.

## Figures and Tables

**Figure 1 cancers-16-01388-f001:**
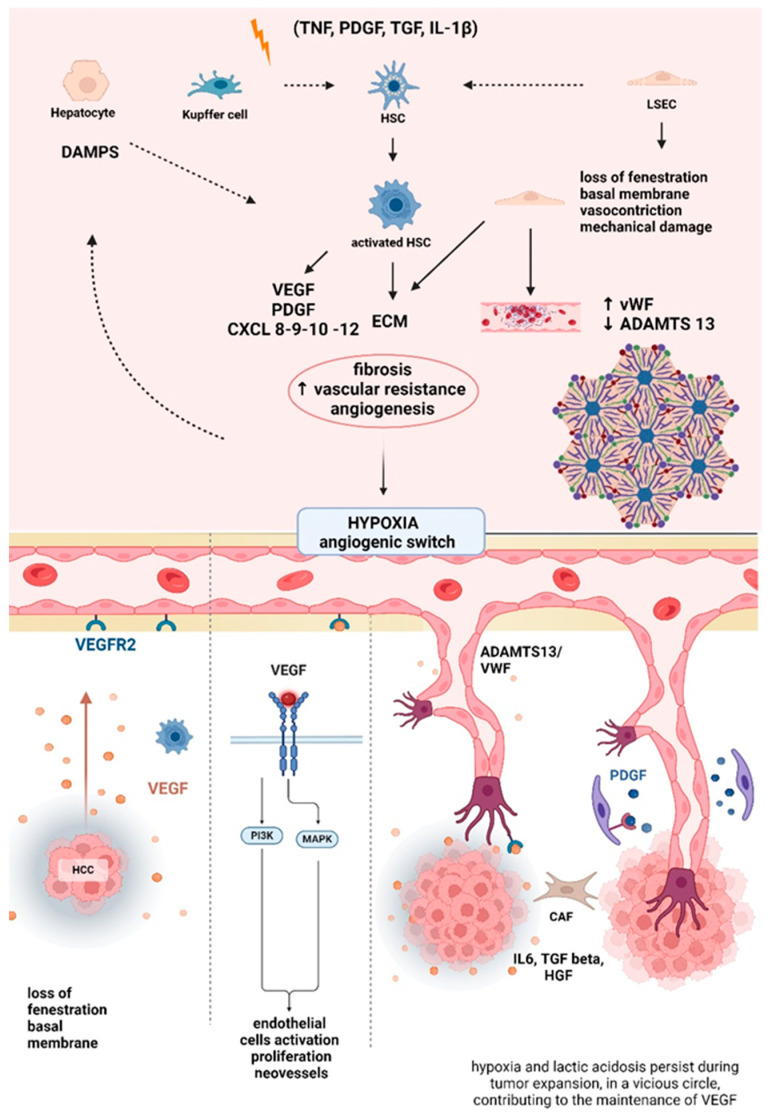
Chronic inflammation, induced by inflammatory cytokines (TNF, PDGF, TNF, and IL-1β), DAMPs, gut dysbiosis, lactic acidosis, and hypoxia, changes the morphology and physiology of liver sinusoids: the capillarization of LSECs (loss of fenestrae and thickening of the basal membrane), which increases intrahepatic portal pressure, and the transformation of HSCs into myofibroblasts are the main changes. In particular, activation of HSCs is stimulated by Kupffer cells and LSECs in a paracrine way and by DAMPs released by hepatocytes. In HSCs, exaggerated production of ECM and lack of ADAMTS-13 are observed. The shear stress induces the exposure of subendothelial prothrombotic substances, such as vWF, that aggregate into multimers and precipitate in the perisinusoidal space, causing platelet aggregation, microthrombus formation, and, in the end, the worsening of liver fibrosis and portal hypertension. HSCs also influence the release of VEGF and its receptors. VEGF/VEGFR interaction produces new vessels that are aberrant and unable to maintain adequate hepatocyte perfusion. Inflammation and hypoxia lead to the so-called “angiogenic switch”, which enhances vessel proliferation; the hypoxic environment induces transcription of HIF-1a, causing the transactivation of target genes such as VEGF. The binding of VEGF/VEGF Receptor isoform 2 activates PI3K/AKT and MAPK, leading to vessel proliferation. Abbreviations: ADAMTS-13: a disintegrin and metalloproteinase with a thrombospondin type 1 motif member 13, DAMPs: damage-associated molecular patterns, ECM: extracellular matrix, LSECs: liver sinusoidal endothelial cells, HCC: hepatocellular carcinoma, HSCs: hepatic stellate cells, HIF-1a: hypoxia inducible factor 1 a, IL-1β interleukin 1 beta, IL-6: interleukin 6, MAPK: mitogen-activated protein kinase, PDGF: platelet-derived growth factor, PI3K phosphatidylinositol 3-kinase/protein kinase B, TGF: transforming growth factor, TNF: tumor necrosis factor, VEGF: vascular endothelial growth factor, VEGFR2: vascular endothelial growth factor receptor 2, vWF: von Willebrand Factor.

**Figure 2 cancers-16-01388-f002:**
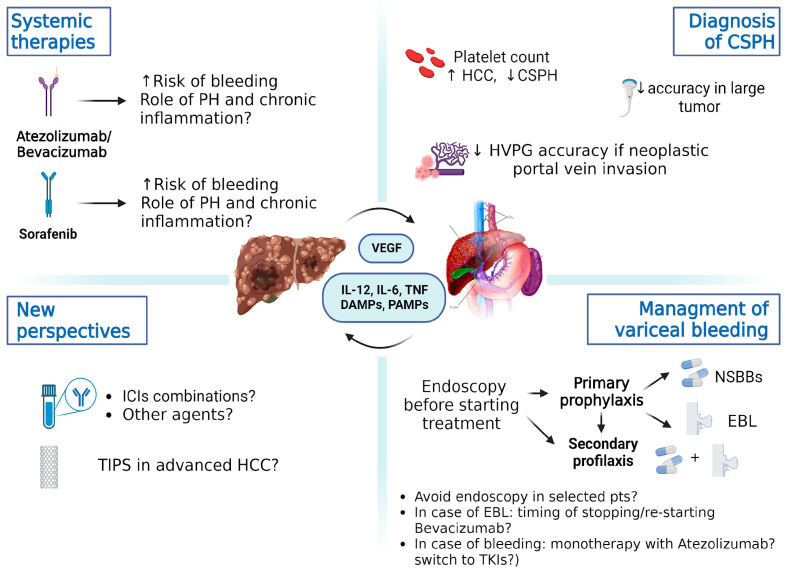
Major complications of liver cirrhosis are portal hypertension and HCC, which share common mechanisms of pathogenesis, including chronic inflammation and angiogenesis. Multiple aspects are involved in the management of portal hypertension. The Baveno VII criteria for CSPH may be affected by variable platelet counts, which may be apparently increased in HCC. Also, transient elastography may reduce the accuracy of liver stiffness estimation and lead to overestimation in the case of large tumors, as well as HVPG. Before starting treatment, endoscopy is mandatory for primary prophylaxis of acute variceal bleeding, according to Baveno VII; NSBBs remain the first choice, given a demonstrated benefit on HCC patient survival; when contraindicated, endoscopic band ligation (EBL) should be performed. Whether endoscopy can be avoided and what the treatment strategy should be after an EBL or bleeding remain open questions. More efforts need to be made to understand the impact of immunotherapy on portal hypertension. In the future, new combinations of immune checkpoint inhibitors or other agents may provide an alternative for patients with impaired liver function or increased risk of bleeding. Abbreviations: CSPH: clinically significant portal hypertension, DAMPs: damage-associated molecular patterns, EBL: endoscopic band ligation, HCC: hepatocellular carcinoma, HVPG: hepatic–venous portal gradient, ICI: immune checkpoint inhibitor, IL-6: interleukin-6, IL-12: interleukin-12, NSBBs: non-selective beta-blockers, PAMPs: pathogen-associated molecular patterns, PH: portal hypertension, TIPS: transjugular intrahepatic portosystemic shunt, TNF: tumor necrosis factor, TKIs: tyrosine kinase inhibitors.

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
