# Peer review of "Management of Portal Hypertension in Patients with Hepatocellular Carcinoma on Systemic Treatment: Current Evidence and Future Perspectives"

_cancers, 2024, doi:10.3390/cancers16071388_

Round 1

Reviewer 1 Report

Comments and Suggestions for Authors

In this review article, de Gaetano and colleagues summarize the current available literature on portal hypertension (PH) and hepatocellular carcinoma (HCC).

The article is comprised of two parts. The first part focuses on the common pathophysiologic and molecular pathways linking together PH and HCC. The second part focuses on the effects of systemic therapy on PH and the resulting implications when making therapeutic decisions.

Although not the first article to address this topic, it provides a very detailed and comprehensive overview of the topic incorporating the latest available evidence.

I only have a few minor suggestions:

1) A figure depicting the intertwined molecular pathways of HCC and PH, particularly those related to angiogenesis, would be particularly helpful

2) Line 193: Even though regorafenib targets VEGF like TKIs, it is a monoclonal antibody rather than a TKI and would therefore mention separately.

3) Lines 282 refer to two different studies, yet the same reference is used. Please double check the accuracy of the references mentioned

4) Line 283 - Please clarify that reduced portal venous flow velocity is suggestive of increased portal congestion

5) Line 314 - The phrase appears incomplete, please revise accordingly

6) Some degree of English language editing is necessary to improve clarity and grammatical accuracy (See below)

Comments on the Quality of English Language

Some language editing is required - mainly grammar/syntax errors, particularly in the simple summary/abstract.

Author Response

We thank the Reviewer for her/his suggestions.

1) "A figure depicting the intertwined molecular pathways of HCC and PH, particularly those related to angiogenesis, would be particularly helpful"

We think this is an useful tool to emphatize the concept and we include a new figure (named figure 1) in the paper.

2) Line 193: "Even though regorafenib targets VEGF like TKIs, it is a monoclonal antibody rather than a TKI and would therefore mention separately."

We highlighted this different mechanism of actions among these drugs in the text (line 231-236)

3) "Lines 282 refer to two different studies, yet the same reference is used. Please double check the accuracy of the references mentioned"

Thank you for the suggestion, we corrected it.

4) "Line 283 - Please clarify that reduced portal venous flow velocity is suggestive of increased portal congestion.

As you pointed out, we provided to modify the text accordingly (lines 328).

5) Line 314 - The phrase appears incomplete, please revise accordingly"

We corrected the mystape.

6) "Some degree of English language editing is necessary to improve clarity and grammatical accuracy (See below). Some language editing is required - mainly grammar/syntax errors, particularly in the simple summary/abstract."

Thank you for your comment, we revised the text and tried to improve the summary and the abstract.

Reviewer 2 Report

Comments and Suggestions for Authors

The review "Management of portal hypertension in patients with hepatocellular carcinoma on systemic treatment: evidence and future perspectives" is devoted to the diagnosis and treatment of hepatocellular carcinoma and systemic problems in this area. It is well known that the use of various tyrosine kinase inhibitors is often quite poorly tolerated and often shows high hepatotoxicity. The drug of choice, according to the authors and literature data, is sorafenib, and its low toxicity with high efficiency is indeed known in clinical practice. The authors are also absolutely right that there is still no consensus on the treatment of this disease, and sometimes, when supervising such clinical cases, treatment regimens are very different. This review helps to take a deeper and broader look at the issue of therapy for hepatocellular carcinoma. The only thing, in the reviewer’s opinion, would be interesting to see the statistics of this disease in different countries of the world. This would give greater relevance to such a theoretical and at the same time clinical meta-analysis. After all, it is known that this disease often occurs almost immediately after infection with hepatitis C. And hepatitis C is a very widespread infection.

I believe that this review deserves to be published in a scientific journal.

Author Response

We thank the Reviewer for her/his suggestion.

We included a brief paragraph about the epidemiology of hepatocellular carcinoma with a focus on the differences in terms of underlying etiologies of liver diseases (lines 44-56). We think this focus will improve our manuscript. 

Reviewer 3 Report

Comments and Suggestions for Authors

This is a detailed review based on current literature data on portal hypertension in patients with 2 hepatocellular carcinomas. This elaboration can be useful for clinicians from the field.

I would only suggest simplifying and additionally explaining description/ specific nomenclature where it can be not fully understood by readers, not from the field.

Author Response

We thank the Reviewer for her/his suggestion. We tried to include some extra explanations about systemic therapy in the main text to favor the Rearders not experts in this field. We also included a new figure to describe the shared mechanisms between portal hypertension and hepatocellular carcinoma, we hope it will help to better understand this intricate relationship.